# Evaluation of the Psychometric Properties of the Portuguese Peabody Developmental Motor Scales-2 Edition: A Study with Children Aged 12 to 48 Months

**DOI:** 10.3390/children8111049

**Published:** 2021-11-13

**Authors:** Miguel Rebelo, João Serrano, Pedro Duarte-Mendes, Diogo Monteiro, Rui Paulo, Daniel Almeida Marinho

**Affiliations:** 1Department of Sport Sciences, University of Beira Interior, 6201-001 Covilhã, Portugal; d.marinho@gmail.com; 2Department of Sports and Well-Being, Polytechnic Institute of Castelo Branco, 6000-266 Castelo Branco, Portugal; j.serrano@ipcb.pt (J.S.); pedromendes@ipcb.pt (P.D.-M.); ruipaulo@ipcb.pt (R.P.); 3Sport, Health & Exercise Research Unit (SHERU), Polytechnic Institute of Castelo Branco, 6000-266 Castelo Branco, Portugal; 4ESECS, Polytechnic of Leiria, 2400-013 Leiria, Portugal; diogo.monteiro@ipleiria.pt; 5Research Centre in Sport Sciences, Health Sciences and Human Development (CIDESD), 6201-001 Covilhã, Portugal

**Keywords:** motor development, child development, motor skills, validation, PDMS-2

## Abstract

The purpose of this study was to examine the psychometric properties of Peabody Developmental Motor Scales II (PDMS-2-Folio and Fewell, 2000) using a Portuguese sample. The validation of the Portuguese version of the PDMS-2 was applied according to the manual, for 392 children, from two institutions, from 12 to 48 months, with an analysis of the internal consistency (α Cronbach), of test–retest reliability (ICC) and construct validity (confirmatory factor analysis). The results of the confirmatory factorial analysis (χ2 = 55.614; df = 4; *p* = 0.06; χ2/df =13.904; SRMR (Standardized Root Mean Square Residual) = 0.065; CFI (Comparative Fit Index) = 0.99, TLI = 0.99, RMSEA (Root Mean Square Error of Approximation) = 0.068) of two factors (Gross Motor and Fine Motor) as the original version but correlated. Most of the subtests had good internal consistency (α = 0.85) and good test–retest stability (ICC = 0.98 to 0.99). The results indicated that the Portuguese version of the PDMS-2 is adequate and valid for assessing global and fine motor skills in children aged 12 to 48 months, and can be used as a reference tool by health and education professionals to assess motor skills and, thus, allowing to detect maladjustments, deficiencies or precocity, so that children can later receive appropriate intervention.

## 1. Introduction

Motor development is a set of change processes that take place throughout life, especially in childhood and adolescence [1]. The same authors also consider that changes in movement and movement patterns change drastically during the first years of life, showing different rhythms of development from child to child, that is, a strong inter-individual variability.

Carvalho [2], states that all children progress according to a typical sequence of stages of development; however, they become unique through socio-cultural influences, experiences and their biological uniqueness [2]. According to these influences, the child reaches all the stages of the development process in the expected periods, but when this does not happen, it is considered a delay in development [2]. The age at which each of these phases must be reached is established according to the average age at which it occurs; there are some variations from one individual to another, and when the maximum period is not reached, the delay is configured. This can occur in one or several areas-for example, global motor, linguistic, social or fine motor skills [2].

In this sense, society and culture can have a profound effect on an individual’s motor behaviors mainly through the practice of physical activity, as well as sociocultural elements such as family, sex, race, religion and nationality, suggesting the need to understand, through a motor assessment with specific tools and instruments [3].

Among the various specific motor assessment tools and instruments described in the literature (Alberta Infant Motor Skills (AIMS), The Gross Motor Function Measure (GMFM), Movement Assessment Battery for Children—Second Edition (MABC-II), Motor Function Measure (MFM), Pediatric Evaluation of Disability Inventory (PEDI) and Test of Gross Motor Development (TGMD-2)), *the Peabody Developmental Motor Scale—second edition (PDMS-2)* [4] is one of the most frequently used instruments in clinical and research settings [5], and unlike other tests, it has the ability to be applied right from the birth of child and allows a complete analysis of global and fine motor skills.

This standardized tool was applied to assess the fine and gross motor skills of children, from birth to 71 months of age, and its normative sample was based on 2003 children residing in 46 states in the United States and in a province of Canada in the year 2000.

In its first edition [6], a *PDMS* was specially designed for the early detection of developmental delays or disturbances. The current revised version [4] has other advantages which specifically allow: the assessment the child’s motor competence in relation to his peers; the identification of motor deficits and imbalances between the fine and gross motor domain; the establishment of individual goals and objectives in clinical and/or educational intervention; and monitorization of the child’s individual development, having the ability to classify the child’s level according to his/her age, on a scale ranging from “very weak” to “very good”. The same authors also highlight the usefulness of PDMS-2, as a research tool, proven with its use in several studies and research projects in the last decade. It can and should be used as a reference for use by educators, doctors and the family to understand how the child is in the motor domains and thus whether there is a need for specific monitoring in any of the motor skills that the child has not yet fully acquired for his or her age.

The usefulness of PDMS-2 as an assessment tool is evident in several studies, which characterized the motor profile of special or clinical populations, such as cerebral palsy, autism, Down syndrome and Hurler syndrome [7,8,9,10,11]. Nevertheless, PDMS-2 has been widely used to analyze the effects of biological (prematurity and malnutrition) and environmental (socioeconomic status, parents’ educational qualifications, quality of the domestic environment, routines established by the family) on child development [12,13,14,15,16,17,18,19,20,21,22,23].

Its acceptance in the scientific community results from the fact that this instrument assesses a multidimensional interpretation of motor behavior, by calculating the following motor composites: Gross Motor Quotient (GMQ), Fine Motor Quotient (FMQ) and the Total Motor Quotient (TMQ), which results from the first two. The segmentation of the TMQ has a very special interest for the differentiation of individual characteristics and, particularly, for the analysis of the effects of intervention programs [24].

According to Folio and Fewell [4], PDMS-2 is a significant improvement on the original version, with regard to the representativeness of the standards and their psychometric properties. In terms of instrument accuracy, the manual reports a good index of internal consistency for each subtest (α = 0.89 to 0.95) and for each motor quotient (0.96 to 0.97), acceptable temporal stability, through the test–retest with an interval of one week (α = 0.73 to 0.96 depending on the age level) and high inter-observer fidelity, which varied between 0.97 to 0.99 for subtests and between 0.96 and 0.98 for the motor quotients. With regard to its construct validity, the two confirmatory factorial studies carried out [6], with two North American gauging sub-samples (up to 11 months and between 12 and 72 months) identified a measurement model consisting of two factors-Fine Motor (FM) and Gross Motor (GM). In another study with Taiwanese children, developed by Chien and Bond [25], when specifically analyzing the dimensionality of the fine motor scale through the Rasch model (1960), concluded that the reduction of some items and the grouping of their two tests (Visual–Motor Integration and Grasping) would make the scale more consistent and more clinically useful. They also found that the 3-point rating scale was not effective and that reducing the scoring category to just a 2-point scale was more representative of skills, and that after modifying the scoring categories, as well the withdrawal of activities, PDMS-2 started to present good psychometric properties. These results show that the validated measurement model, for the North American sample, may not be suitable or identical for another distinct population, so it is prudent to proceed with its cross-cultural adaptation before application [25]. Regarding concurrent validity, the authors of the instrument [4] concluded that as PDMS-2, presenting a high correlation with its original version (α = 0.84 and 0.91 respectively for GMQ and FMQ) and with the *Mullen Scales of Early Learning* (Mullen, 1995) (α = 0.86 and 0.80 respectively for GMQ and FMQ).

Bean et al. [26], when assessing children at risk of development, aged between two and fifteen months, registered good rates of internal consistency (α = 0.90 and 0.97) between the results of three subtests (Reflexes, Stationary and Locomotion) the PDMS-2 gross motor scale and the total motor quotient of the *Alberta Infant Motor Scale* [27]. In turn, Connolly et al. [28], analyzed the concurrent validity between PDMS-2 and *Bayley Scales of Infant Development II* (BSID-II) with 12-month-old children. The results showed a low correlation between the standard values of the motor quotients the PDMS-II and the Psychomotor Development Index of the BSID-II (α = 0.30; 0.22 and 0.32 respectively for GMQ, FMQ and TMQ). Only a high correlation was found in the values referring to age for the Locomotion test (α = 0.71, *p* < 0.05). Based on these results, Bean et al. [26] advise prudence in the interpretation of standardized values or values referring to age, when making clinical decisions based on a single assessment instrument.

The sensitivity of the instrument was confirmed by the authors of the scales [4], depending on age, sex, ethnicity (European Americans, African Americans and Hispanic Americans) and motor and mental deficits. Additionally, Wang, Liao and Hsieh [29] also tried to test the sensitivity in a sample of children with cerebral palsy, aged between two and five years, the results suggested a sensitivity to developmental changes for an interval of six months. This appears to be an important improvement on the revised version, since Palisano et al. [30] had reported that the gross motor scale, of the original version of the PDMS, was not able to detect changes in the motor development of children with cerebral palsy, in an interval of six months.

Despite all the metric evidence, some authors [31,32,33,34] have warned that the application of PDMS-2, and particularly the interpretation of its standardized values, for certain special/clinical groups or in contexts culturally different from those for which the instrument was originally developed, should be developed with some caution, and recommend a cross-cultural adaptation and validation of the instrument to the population concerned. Regarding the reliability of the instrument for the Portuguese population, Saraiva et al. [24], reported in their adaptation and validation study, that most subtests had a good index of internal consistency (α = 0.76 to 0.95) and good test–retest stability (ICC = 0.85 to 0.95), concluding that the results indicate that the Portuguese version of PDMS-2 is an accurate and valid instrument for assessing the gross and fine motor skills of Portuguese children of pre-school age (from 36 to 72 months). However, Saraiva et al. [24] suggest that it is essential to replicate the same study in different age groups, highlighting the age range from 0 to 36 months.

Thus, as an indicator and support for the assessment of motor skills by health and education professionals, it is pertinent to verify whether the PDMS-2 scales are suitable for the Portuguese population aged 12 to 48 months, so that it can be used as assessment instrument that allows for the detection of maladjustments, deficiencies or precociousness, and the child can later receive the appropriate intervention.

In this sense, the objective of the study was to analyze the psychometric properties of the Portuguese version of the *Peabody Developmental Motor Scales II* (PDMS-2) for the Portuguese population from 12 to 48 months.

According to the existing literature [3,24,35], it is expected that the Portuguese version of PDMS-2 will present psychometric properties similar to those of the original version in terms of its characterization, precision and theoretical construct, and that it will be an accurate and valid instrument to assess the gross and fine motor skills of Portuguese children aged 12 at 48 months.

## 2. Materials and Methods

### 2.1. Participants

This study fits into a quantitative correlational typology, being a cross-sectional study. As for the nature of the sample, we can say that it is intentional, for convenience, since it is appropriate to the type of study we intend to carry out and we consider it to be non-probabilistic, since it was selected by the researcher’s subjective criteria and according to the purpose of the study [36].

The study took place in nurseries and kindergartens and consisted of a total of 392 subjects of both sex with ages (M = 29.86 ± SD = 8.79 months) between 12 and 48 months (Male, *n* = 199, M = 29.94 ± SD = 8.73 months; Female, *n* = 193, M = 29.78 ± SD = 8.87 months) from public and private institutions, from urban, semi-urban and rural areas in the district of Castelo Branco, Portugal (information on the type of residence was categorized through the anamnesis form applied to each child, these data were reported by the fact that the original authors of the PDMS-2 validation consider this information relevant in the characterization of the sample).

For the selection of the sample, the following inclusion criteria were defined: children aged between 12 and 48 months; of Portuguese nationality; and as an exclusion criteria: children diagnosed with learning difficulties and/or developmental impairments and/or children with some type of diagnosed disability, respectively (this exclusion factor is due to the fact that the authors of PDMS-2 and the authors first validation for the Portuguese population to use only children without any associated condition).

All ethical principles, international norms and standards regarding the Helsinki Declaration and the Convention on Human Rights and Biomedicine were followed, respected and preserved [37]. This project was approved by the Ethics Committee of the institution where the authors carry out their research.

The Table 1 summarizes the main sociodemographic characteristics of the sample by age group.

### 2.2. Instruments

Studies with PDMS-2 were authorized by the publisher PRO-ED, Incorporated., from Austin, Texas. In this process, we tried to follow the methodological procedures recommended in the specific literature regarding the adaptation of an evaluation instrument [38,39] since all the translation processes of the instrument had already been carried out by Saraiva, Rodrigues, and Barreiros [24], having adopted all the procedures that aimed to ensure linguistic, conceptual, operational and psychometric equivalence between the translated Portuguese version and the original version.

Every version of the PDMS-2 was translated by Saraiva et al. [24] but it was only applied to children aged 36 to 71 months. Thus, based on the Portuguese version of the PDMS-2, it was applied to 392 children from 12 to 48 months, age group not applied by the authors of the first adaptation and validation for the Portuguese population (from 12 to 36 months).

The composite structure of the PDMS-2 includes five subtests distributed over two motor components/scales: gross motor skills and fine motor skills. Its results are expressed in three domains of motor behavior: Fine Motor Quotient (FMQ), Gross Motor Quotient (GMQ) and total motor quotient (TMQ), the latter resulting from the first two. The FMQ is found by the sum of two sets of subtests, namely, Grasping and Visual–Motor Integration, while for the GMQ three are used: Stationary, Locomotion and Object Manipulation (the latter is replaced by the Reflexes subtest for children up to eleven months old). Each of these subtests consists of items (motor tasks) adjusted for age and placed in an increasing sequence of difficulties. The child starts the test on a specific item, according to his age, and continues in the sequence until three consecutive runs fail.

Each item is classified according to an evaluation scale, of three values: 0 = does not perform, 1 = minimum proficiency, 2 = optimal proficiency. The sum value of all items, in each of the subtests, is located in the reference table for age, resulting in a standardized value and a percentage value that can be compared between ages. Subsequently, the sum of the standardized values of the grouped subtests, allows obtaining the total motor quotient, fine or gross, by consulting a second table. The standardized scales for the North American child population establish the average value of 10 points (±3) for each test and the average value of 100 points (±15) for the motor quotients. Standardized values can also be converted into a qualitative classification of seven categories (between “Very Good” and “Very Weak”) [24].

### 2.3. Procedures

After approval by the institutions, the necessary authorizations were obtained, and informed consent was also requested from the guardians, whom all the procedures and the purpose of the study were explained.

The children were assessed individually, by two specially trained investigators, experienced in the health and sport field and familiarized with the PDMS-2 application protocol, with a percentage of 90% interobserver agreements reached, in the item-by-item quotation before data collection. The administration of the scales took about 30 to 45 min per child, depending on the age group. The evaluation took place in an empty kindergarten room, in a non-intrusive environment and adequate to the protocol described in the PDMS-2 manual [4]. Regarding the test–retest reliability in a subsample, in addition to using exactly the same application and data collection procedures, this was calculated according to approximately 10% [40,41,42,43,44,45] of the total of our sample (30 children), presenting a time interval between the test and the re-test of seven days, having been carried out under exactly the same conditions and by the same investigator. The raw scores, obtained in the subtests, were converted into standard scores, and the respective gross, fine and total motor quotients were calculated, based on the values referenced in the manual.

### 2.4. Statistical Analysis

Initially, measures of central tendency (mean) and measures of dispersion (standard deviation) were analyzed and the comparison between age groups with One-Way Anova, as well as bivariate correlations between all variables analyzed. In a second instance, in order to respond to the objective of the present study, a Confirmatory Factor Analysis was performed (CFA). To carry out the CFA, the recommendation of a ratio was considered 10:1 (i.e., number of subjects for each parameter to be estimated in the model) suggested by several authors [40,41,42,43,44,45]. Data analysis was performed according to the guidelines and recommendations of several authors [40,42,44]: in addition to the maximum likelihood estimation method (MLE), through the chi-square test (χ^2^), the respective degrees of freedom (*df*) and the level of significance (*p*), the following adjustment quality indices were also used: *Standardized Root Mean Square Residual* (SRMR), *Comparative Fit Index* (CFI), *Non-Normed Fit Index* (NNFI), *Root Mean Square Error of Approximation* (RMSEA) and the respective confidence interval (90% CI). In the present study, for the referred indices, cut-off values were adopted, suggested by Byrne [40,41]; Hair et al. [42,43] and Marsh, Hau and Wen [46]: SRMR ≤ 0.08, CFI and NNFI ≥ 0.90 and RMSEA ≤ 0.08. Convergent validity was also analyzed (to check if the items are related to the respective factor), through average variance extracted (AVE) calculation, considering adjusted values of AVE ≥ 50 [12,13,17] and the discriminant validity was verified (to verify that the factors are sufficiently distinct from each other), considering adjusted with the square of the correlation between the factors is less than the value of AVE on both factors [42], as well as, the composite reliability (CR), to evaluate the internal consistency of the factors, adopting as cut-off values ≥ 0.70 de CR, as suggested by Hair et al. [42]. The data were analyzed using the software AMOS 23.0 and SPSS software v. 25.0 [43], and the significance level was set at *p* < 0.001 to reject null hypothesis.

## 3. Results

### 3.1. Preliminary Analysis

A preliminary analysis of the data revealed that there were no missing values, univariate and multivariate outliers. The results also revealed that there were no violations to the univariate normal distribution, since the values of Skewness and Kurtosis were included between −2 +2 and −7 +7 [40,41], respectively. Notwithstanding, Mardia’s coefficient for multivariate kurtosis revealed a normal multivariate distribution (2.618), this value is lower than that recommended in the literature (5.0), as suggested by several authors [42,44], so no measures were taken against multivariate non-normality [41].

Table 2 shows the descriptive analysis of the raw scores and the sub-tests according to the age group. In a brief analysis of the data, it appears that the average values obtained in each subtest, show an increase over the age groups, which shows the characterization of the sample. The variability of the results is also visible in all subtests, except in the Grasping test, which only records a standard deviation of 1.7 and 2.7 in the 2- and 3-year age groups, respectively.

### 3.2. Test-Retest Reliability

The Precision Study of the instrument, included the analysis of the internal consistency of the subtests and the test–retest reliability of the results obtained, by 30 children in our sample after a retest in an interval of 7 days. Table 3 shows the results for these precision parameters.

By interpreting Cronbach’s alpha values, it can be said that most subtests obtained a good internal consistency index, oscillating between 0.84 and 0.97. The subtests recorded excellent values for internal consistency of Locomotion (α = 0.97), Object Manipulation (α = 0.93), and Visual–Motor Integration (α = 0.96), and good values of internal consistency for Stationary (α = 0.86) and Grasping (α = 0.84). With regard to the test–retest reliability estimated through the intraclass correlation coefficient (ICC), it can be seen in Table 3 that the values varied between 0.98 and 0.99.

### 3.3. Construct Validity

The PDMS-2 factorial model tested for the Portuguese sample was identical to the one originally proposed by the authors (Figure 1); that is, a model of two latent factors, but that in our models these two factors are correlated (gross motor skills and fine motor skills) defined respectively by three (Stationary, Locomotion and Object Manipulation) of gross motor skills and two (Grasping and Visual–Motor Integration) items of fine motor skills. Its adjustment was tested through a confirmatory factor analysis.

By analyzing Figure 1, it appears that all items have a factor weight ≥ 0.50 (all statistically significant, *p* < *0*.05), therefore explaining at least 25% of the latency factor variance [42,44]. Additionally, the measurement model revealed good values of convergent validity, since the AVE of both factors was higher than 0.50 [47,48]. However, the instrument revealed problems of discriminant validity, since the square of the correlation between the factors (r^2^ = 0.96) was higher than the AVE value on both factors (0.82; 0.80), for the gross motor and fine motor factors, respectively. Finally, the instrument revealed good values of internal consistency, since the values of composite reliability of both factors were higher than 0.70 [42,44,49].

Based on Table 4, it turns out that the measurement model of the analyzed instrument presented a good adjustment to the data, according to the cut-off values adopted in the methodology [13,40,47].

## 4. Discussion

The aim of the present study was to analyze the psychometric properties of the Portuguese version of the Peabody Developmental Motor Scales II (PDMS-2) for the Portuguese population from 12 to 48 months. In the preliminary analysis, the data revealed that there were no missing values, nor univariate and multivariate outliers, and that there were no violations of the univariate normal distribution. The results showed in the study of the accuracy of the instrument, that the Portuguese version PDMS-2 revealed, on the whole, very satisfactory indexes. As well as presenting good indexes of adjustment in the confirmatory factor analysis.

### 4.1. Preliminary Analysis

In the sample characterization study, we verified an excellent adaptation and performance of the studied children in relation to the norms of PDMS-2, and there was confirmation that the subtests of Visual–Motor Integration, Stationary, Locomotion and Object Manipulation were able to discriminate the motor performance of Portuguese children between the ages of 12 and 48 months. The average scores (raw scores) of these subtests, observed through the descriptive and statistical analysis of the sample data, registered an increase over age and their respective standard deviations (greater than 2.3) prove the variability of the results obtained in the study sample as had already happened in the study by Saraiva et al. [24].

The same cannot be said in relation to the Grasping subtest, since the raw scores achieved by the two-year-old children only showed a variability of 1.7. In fact, this seems to be a real limitation of the instrument, regardless of the population concerned, as this result was also reported for Taiwanese children [25], Flemish [50] and Portuguese [24] in preschool age. From a clinical point of view, Van Hartingsveldt, Cup and Oostendorp [34], to when evaluating 18 Dutch children aged 4 and 5, they concluded that the fine motor skills scale of the PDMS-2 demonstrated a lower sensitivity to discriminate children with slight fine motor problems compared to the Movement Assessment Battery for Children.

### 4.2. Test–Retest Reliability

Regarding the test–retest reliability of the instrument, it can be inferred that the Portuguese version PDMS-2 showed, on the whole, very satisfactory and comparable indexes to the original version, such as Saraiva et. al. [24] had already achieved but at different ages. Most subtests reached a Cronbach’s alpha value substantially higher than the cut-off point of 0.70 proposed by Hair et al. [42] and as already achieved in the study by Saraiva et. al. [24] but in this study only in the Stationary, Locomotion and Object Manipulation subtests. In terms of test–retest accuracy, it was found that the subsample of 30 Portuguese children, in an interval of 7 days between the two applications, also registered high stability coefficients (ICC ≥ 0.98) in all subtests. These PDMS-2 accuracy indexes have been confirmed in other psychometric studies [10,24,29,34,51,52].

### 4.3. Construct Validity

As for the construct validity, the results of the confirmatory factor analysis support that the Portuguese version PDMS-2 presents a model of two factors: gross motor and fine motor, just like the original version proposed by Folio and Fewell [4], but in our model these two factors are correlated. The adjustment indexes of the Portuguese model from 12 to 48 months (χ^2^ = 55.614, df = 4, *p* = 0.06) and SRMR = 0.065; TLI= 0.992; CFI = 0.998; and RMSEA = 0.068 90%IC= 0.000–0.138) were very similar to the Portuguese model from 36 to 71 months for Saraiva, et. al. [24], (S-Bχ2 = 3.3, *p* = 0.349; CFI = 1.0; NFI = 0.99; NNFI = 0.99; RMSEA = 0.013) and those of the original North American version (TLI = 0.96; RMSEA = 0.08). Finally, to be highlighted is the fact that the Portuguese factorial structure from 12 to 48 months registered higher values of item-factor saturation (α = 0.81 to 0.99), compared to the Portuguese factorial structure from 36 to 71 months (α = 0.67 to 0.95) and the original American structure (α = 0.54 a 0.89) respectively, which demonstrates a greater relevance of the values of the items (subtests) in the determination of the respective latent factors (gross motor and fine motor).

## 5. Study Limitations

Among the limitations of this study is the fact that the research design is cross-sectional, as well as the application of the instrument in children of such a premature and time-consuming age group. These can influence the results obtained, since sometimes children may not perform an exercise to perfection, not because they do not know but because they do not trust the evaluator. Another limitation is the fact that children with disabilities were not included in the study as this instrument also allows its application in this population.

## 6. Conclusions

In short, we can conclude that the Portuguese version PDMS-2 proved to be an accurate and valid instrument to assess the fine and global motor skills of Portuguese children aged 12 to 48 months. The different empirical analyzes carried out within the scope of this study showed that the Portuguese version has psychometric characteristics similar to those of the original version in terms of its characterization, precision and theoretical construction, but it is advisable that further research be conducted to establish other aspects of validity and reliability, for its use and credibility in the national context, and very importantly in the educational, clinical and scientific areas.

It is suggested that, in future studies, advantage should be taken of the instrument’s potential, that the PDMS-2 validation and measurement process be consolidated with the replication of the same study, using more psychometric properties of the COSMIN (COnsensus-based Standards for the selection of health Measurement INstruments) categories for the Portuguese population, as well as validation for unexplored age groups (from zero to 1 year) would be essential.

## 7. Practical Applications

The results obtained by our study suggest that the PDMS-2 scales can be used as a reference instrument by health and education professionals, as an indicator and support for the assessment of motor skills, thus having an assessment instrument that allows for the detection of maladjustments, deficiencies or precociousness, so that the child can later receive the appropriate intervention. However, it is important to emphasize that the acquisition of skills is not directly and intrinsically linked to time, but to the development process that is unique to each human being [53].

## Figures and Tables

**Figure 1 children-08-01049-f001:**
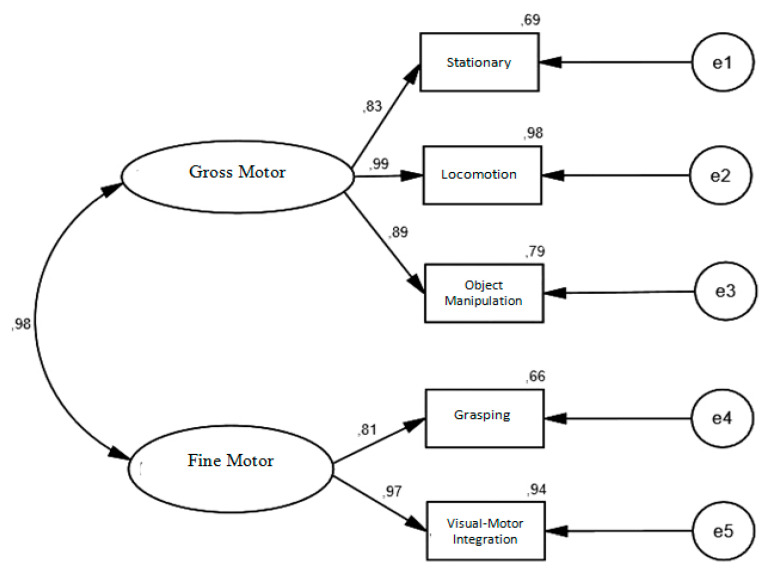
Factorial structure of the PDMS-2 measurement model for the Portuguese sample aged between 12 and 48 months.

**Table 1 children-08-01049-t001:** Sociodemographic characterization of the 392 Portuguese children.

		12–23 Months (*n* = 96)	24–35 Months (*n* = 153)	36–47 Months (*n* = 143)	Totals (*n* = 392)
Age (Average ± SD)		18.66 ± 3.91	28.07 ± 3.35	39.30 ± 3.55	29.86 ± 8.79
Sex (%)	Male	44 (45.8)	79 (51.6)	76 (51.6)	199 (50.8)
	Female	52 (54.2)	74 (48.4)	67 (48.4)	193 (49.2)
Residence (%)	Urban	60 (62.5)	132 (86.3)	125 (87.4)	317 (80.9)
	SemiUrban	21 (15.6)	3 (2.0)	16 (11.2)	40 (10.2)
	Rural	15 (21.9)	18 (11.8)	2 (1.4)	35(8.9)

**Table 2 children-08-01049-t002:** Mean, standard deviation (SD), minimum-maximum value, amplitude of the Raw scores and comparison between groups, obtained in the sub-tests by age group.

Subtests		12–23 Months (*n* = 96)	24–35 Months (*n* = 153)	36–48 Months (*n* = 143)	*Sig.*
Stationary	Mean (SD)	39.4 (2.3)	43.3 (2.8)	48.9 (3.8)	
	Min–Max *	36–46 *	38–51 *	41–56 *	<0.001
	Amplitude	10	13	15	
Locomotion	Mean (SD)	81.4 (13.5)	111.6 (13.5)	137.8 (10.1)	
	Min–Max *	50–100 *	87–136 *	115–158 *	<0.001
	Amplitude	50	49	43	
Object Manipulation	Mean (SD)	13.2 (5.5)	21.1 (5.8)	30.0 (5.1)	
	Min–Max *	1–26 *	8–35 *	20–42 *	<0.001
	Amplitude	25	27	22	
Grasping	Mean (SD)	39.9 (3.8)	43.3 (1.7)	48.2 (2.7)	
	Min–Max *	34–46 *	40–47*	42–52 *	<0.001
	Amplitude	12	7	10	
Visual–Motor Integration	Mean (SD)	79.7 (9.83)	99.9 (11.1)	121 (8.8)	
	Min–Max *	53–96 *	79–122 *	109–139 *	<0.001
	Amplitude	43	43	30	

Note. * Maximum possible score: Stationary (*n* = 60); Locomotion (*n* = 178); Object Manipulation (*n* = 48); Grasping (*n* = 52); Visual–Motor Integration (*n* = 144).

**Table 3 children-08-01049-t003:** Internal consistency and test–retest reliability of subtests.

Subtest	Internal Consistency (α Cronbach) *n* = 392	Test–Retest Reliability (ICC) *n* = 30
Stationary	0.86	0.98
Locomotion	0.97	0.99
Object Manipulation	0.93	0.98
Grasping	0.84	0.99
Visual–Motor Integration	0.96	0.99
Gross Motricity	0.71	0.75
Fine Motricity	0.69	0.71
Total Motricity	0.85	0.87

**Table 4 children-08-01049-t004:** Adequacy indexes of the model tested between 12 and 47 months.

Index	χ^2^	χ^2^/DF	SRMR	TLI	CFI	RMSEA	90%IC
M 1	55.614	13.904	0.065	0.992	0.998	0.68	0.000–0.138

## Data Availability

The datasets generated and/or analyzed during the current study are not publicly available due [REASON WHY DATA ARE NOT PUBLIC] but are available from the corresponding author on reasonable request.

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
