# Peer review of "Evaluation of the Psychometric Properties of the Portuguese Peabody Developmental Motor Scales-2 Edition: A Study with Children Aged 12 to 48 Months"

_children, 2021, doi:10.3390/children8111049_

Round 1
Reviewer 1 Report
Is a great manuscript and very well done. I do have suggestions for the authors to improve the readability of the manuscript.
Introduction: Line 45 the authors list examples of development and list 'fine motor' twice. Lines 46-50 is one long run-on sentence that should be broken into multiple sentences to enhance readability. Lines 52-58 the authors list the different motor tests available, which is great. However, this very long sentence is confusing and they should break this up. Overall the introduction did a great job explaining the topic at hand and cited past studies.
Methods: line 169 the authors use the term 'gender' when it is more appropriate to use 'sex' for describing biological sex. This should be used throughout the document. On table 1 there is no need for the 'Note n=392' under the table.
Results: Line 311 says 'etherfore'.
Overall, I think the study is very novel and is in great condition.
Author Response
Response to Reviewer 1 Comments
Point 1: Introduction:
Line 45 the authors list examples of development and list 'fine motor' twice. Lines 46-50 is one long run-on sentence that should be broken into multiple sentences to enhance readability. Lines 52-58 the authors list the different motor tests available, which is great. However, this very long sentence is confusing and they should break this up. Overall the introduction did a great job explaining the topic at hand and cited past studies.
Response 1: Thanks for the sugestion. We agree and carry out all proposed changes.
Action in the corrected version of the manuscript:
In line 45, there was a writing error and replaced "fine motor" with "global motor".
In line 46-50, the sentence has been simplified for a better understanding: " In this sense, society and culture can have a profound effect on an individual's motor behaviors mainly through the practice of physical activity, as well as sociocultural elements such as family, gender, race, religion and nationality, sugsuggesting itself the need to understand, through a motor assessment with specific tools and instruments [3].”
In line 52-58, as suggested, the sentence has been split for a better understanding of the readers.
Point 2: Methods:
line 169 the authors use the term 'gender' when it is more appropriate to use 'sex' for describing biological sex. This should be used throughout the document. On table 1 there is no need for the 'Note n=392' under the table.
Response 2: Thank you for the suggestion. We appreciate the suggestion and it has been changed throughout the manuscript "gender" to "sex". As well as removing "'Note n=392'" from Table 1.
Point 3: Results:
Line 311 says 'etherfore'
Response 3: Thank you for the suggestion. As proposed, was replaced "etherefore" by "etherfore ", in line 311 of the manuscript.
Reviewer 2 Report
The study examined the psychometric properties of Peabody Developmental Motor Scales using a Portuguese sample. The participants were 392 children from 12 to 48 months. The results indicate that Portuguese version of this scale is adequate for assessing global and fine motor and gross motor skills in this range of age. Then, this scale would allow detecting difficulties and motor problems in this population to provide an adequate intervention. This is an interesting article that studies an area of great need of more research. However, there are significant observations that I consider should be addressed before accepting the manuscript for publication:
Materials and Methods
In the Ln. 169 the authors must point the mean value with the letter “M” and the standard deviation with the acronymous “SD”. These values are enclosed in parentheses.
They describe the age of the participants between 12 and 47 moths. Is it 47 or 48 moths? Clear out.
There are some spelling and grammar errors in this section.
In the note in Table 1, the authors should specify what “n” means. In addition, they have omitted the letter “N” in the “residence” row.
Instruments
The authors have repeated the following idea “Despite all the metric evidence, some authors [31-34] have warned that the application of PDMS-2, and particularly the interpretation of its standardized values, for certain special / clinical groups or in contexts culturally different from those for which the instrument was originally developed, should be developed with some caution, and recommend a cross-cultural adaptation and validation of the instrument to the population concerned”.
In addition, they repeat information in this same section that is already included in the introduction section. The description of the test is done either in the introduction or in the instruments section:
“The composite structure of the PDMS-2 includes five subtests distributed over two motor components / scales: gross motor skills and fine motor skills. Its results are expressed in three domains of motor behavior: Fine Motor Quotient (FMQ), Gross Motor Quotient (GMQ) and total motor quotient (TMQ), the latter resulting from the first two. The FMQ is found by the sum of two sets of subtests, namely, Grasping and Visual-Motor Integration, while for the GMQ three are used: Stationary, Locomotion and Object Manipulation (the latter is replaced by the Reflexes subtest for children up to eleven months old)”.
The total score for each of the subscales is collected in Table 2, it is necessary that the authors put this information in the last paragraph of the instruments section.
It must add the references of AMOS and SPSS software.
I suggest using the same criteria: put the zero before the value decimal or not put it. There are results that have it and others that do not.
Results
The authors explain that “The variability of the results is also visible in all subtests, except in the Grasping test, which only records a standard deviation of 1.7 and 2.7 in the two and three year age groups, respectively”. However, the variability in Stationary test in the age groups of one and two years is also small. Authors should highlight these scores and discuss these results in the results section.
I suggest putting a line to separate the values of subtest of the scores gross motricity, fine motricity and total motricity in Table 3.
The word “Figure” must be capitalized.
I do not understand the meaning or what it refers to “The tables may have a note. ICC> 0.98 "
There must be a mistake in the references from the line 322.
REFERENCES
The style of the manuscript references does not conform to the standards of the journal. Authors should review the references section.
Throughout the text they write the last name of the referenced authors accompanied by the number they occupy in the reference list. In some cases, such as ln. 141, it would not be necessary to put the authors' last name, only the reference number. In other cases, they should replace the citation with the number that it occupies in the list of references.
Author Response
Response to Reviewer 2 Comments
1- Materials and Methods:
In the Ln. 169 the authors must point the mean value with the letter “M” and the standard deviation with the acronymous “SD”. These values are enclosed in parentheses.
They describe the age of the participants between 12 and 47 moths. Is it 47 or 48 moths? Clear out.
There are some spelling and grammar errors in this section.
In the note in Table 1, the authors should specify what “n” means. In addition, they have omitted the letter “N” in the “residence” row.
Response 1: Thank you very much for your comment and suggestions.
In line 169, as proposed, the abbreviations “M” and “SD” were introduced.
We appreciate the call to attention and the error was corrected with the children's months.
On the recommendation of reviewer 1, the grade and the letter "n" in table 1 were removed.
2- Instruments:
The authors have repeated the following idea “Despite all the metric evidence, some authors [31-34] have warned that the application of PDMS-2, and particularly the interpretation of its standardized values, for certain special / clinical groups or in contexts culturally different from those for which the instrument was originally developed, should be developed with some caution, and recommend a cross-cultural adaptation and validation of the instrument to the population concerned”.
In addition, they repeat information in this same section that is already included in the introduction section. The description of the test is done either in the introduction or in the instruments section:
“The composite structure of the PDMS-2 includes five subtests distributed over two motor components / scales: gross motor skills and fine motor skills. Its results are expressed in three domains of motor behavior: Fine Motor Quotient (FMQ), Gross Motor Quotient (GMQ) and total motor quotient (TMQ), the latter resulting from the first two. The FMQ is found by the sum of two sets of subtests, namely, Grasping and Visual-Motor Integration, while for the GMQ three are used: Stationary, Locomotion and Object Manipulation (the latter is replaced by the Reflexes subtest for children up to eleven months old)”.
Response: Thank you for the suggestion. We appreciate and accept your suggestion. However, reviewer 1 initially proposed to introduce some information in the introduction about the instrument. However, and agreeing with your suggestion, and in order not to repeat the same information in the introduction and in the instruments, some information about the instrument was removed in the introduction, as is the example in line 97 of the manuscript.
The total score for each of the subscales is collected in Table 2, it is necessary that the authors put this information in the last paragraph of the instruments section.
Response:We appreciate and understand your suggestion to include information at the end of the paragraph regarding the total score for each of the subscales. However this is taken from a reference table for the specific age of each child, but it is not table 2, but a specific reference table. Due to the rights of the authors of the instrument, this table cannot be reported. In this sense, we added this information to the manuscript (Line 216).
It must add the references of AMOS and SPSS software.
Response:We appreciate the suggestion and have added references to AMOS and SPSS.
I suggest using the same criteria: put the zero before the value decimal or not put it. There are results that have it and others that do not.
Response: Thank you for your attention and the same criterion was always used, with "0" being removed from the entire manuscript before the decimal value.
3- Results:
The authors explain that “The variability of the results is also visible in all subtests, except in the Grasping test, which only records a standard deviation of 1.7 and 2.7 in the two and three year age groups, respectively”. However, the variability in Stationary test in the age groups of one and two years is also small. Authors should highlight these scores and discuss these results in the results section.
Response: We appreciate and accept your suggestion. These scores were highlighted and discussed in the results section line 342 of the Manuscript.
I suggest putting a line to separate the values of subtest of the scores gross motricity, fine motricity and total motricity in Table 3.
Response: Thank you for the suggestion. As suggested, a line was placed separating the gross, fine and total motor subtests.
The word “Figure” must be capitalized.
Response: Thank you for the suggestion. It was corrected and put the word "figure" in capital letters.
I do not understand the meaning or what it refers to “The tables may have a note. ICC> 0.98 "
Response: Thank you very much for the reminder. However this note had not been placed, it must have been an editing error, however this note was removed.
There must be a mistake in the references from the line 322.
Response: Thank you very much for the suggestion, this error has been duly corrected in the manuscript.
4- References:
The style of the manuscript references does not conform to the standards of the journal. Authors should review the references section.
Throughout the text they write the last name of the referenced authors accompanied by the number they occupy in the reference list. In some cases, such as ln. 141, it would not be necessary to put the authors' last name, only the reference number. In other cases, they should replace the citation with the number that it occupies in the list of references.~
Response 4: Thank you very much for the reminder and suggestion. All references were placed in accordance with the journal's standards.
Regarding the text, sometimes the authors' surnames are shown followed by the reference number, when they are cited in the first person, as referred to in the journal's rules followed by the editor's recommendations.
Round 2
Reviewer 2 Report
When I proposed capitalizing the word "Figure", I was referring to the body of the manuscript and not the title of the figure that was spelled correctly. The first letter of the word Figure must be capitalized. The same in the word "Table".